# Glutamate Dehydrogenase Functions in Glutamic Acid Metabolism and Stress Resistance in *Pyropia haitanensis*

**DOI:** 10.3390/molecules26226793

**Published:** 2021-11-10

**Authors:** Shuang Li, Zhanru Shao, Chang Lu, Jianting Yao, Yongdong Zhou, Delin Duan

**Affiliations:** 1CAS and Shandong Province Key Laboratory of Experimental Marine Biology, Center for Ocean Mega-Science, Institute of Oceanology, Chinese Academy of Sciences, Qingdao 266071, China; ouclishuang@163.com (S.L.); luchang@qdio.ac.cn (C.L.); yaojianting@qdio.ac.cn (J.Y.); 2Laboratory for Marine Biology and Biotechnology, Qingdao National Laboratory for Marine Science and Technology, Qingdao 266071, China; 3University of Chinese Academy of Sciences, Beijing 100049, China; 4Yancheng Hairui Food Co., Ltd., Dafeng 224005, China; 13705112139@139.com

**Keywords:** heterologous expression, protein purification, enzyme activity assay, site-directed mutagenesis, abiotic stresses

## Abstract

*Pyropia haitanensis* is an important laver species in China. Its quality traits are closely related to the content of glutamic acid. Glutamate dehydrogenase (GDH) is a crucial enzyme in the glutamic acid metabolism. In this study, two *GDH* genes from *P. haitanensis*, *PhGDH*1 and *PhGDH*2, were cloned and successfully expressed in *Escherichia coli*. The in vitro enzyme activity assay demonstrated that the catalytic activity of PhGDHs is mainly in the direction of ammonium assimilation. The measured *K_m_* values of PhGDH1 for NADH, (NH_4_)_2_SO_4_, and α-oxoglutarate were 0.12, 4.99, and 0.16 mM, respectively, while the corresponding *K_m_* values of PhGDH2 were 0.02, 3.98, and 0.104 mM, respectively. Site-directed mutagenesis results showed that Gly^193^ and Thr^361^ were important catalytic residues for PhGDH2. Moreover, expression levels of both PhGDHs were significantly increased under abiotic stresses. These results suggest that PhGDHs can convert α-oxoglutarate to glutamic acid, and enhance the flavor and stress resistance of *P. haitanensis*.

## 1. Introduction

*Pyropia* is an important genus of red algae. According to the Food and Agriculture Organization of the United Nations (FAO), the laver production was about 1.8 million tons (fresh weight) in 2016, with a commercial value of more than USD 1.5 billion [1]. In China, naturally there are about 22 *Pyropia* species, among which the two most important species are *P. yezoensis* and *P. haitanensis* [2]. *P. haitanensis* is cultivated on a large scale and shows the highest yield among all laver species [1]. It is extensively distributed in the coastal areas of Guangdong, Fujian, Zhejiang, and Jiangsu provinces [3]. At present, the annual output of *P. haitanensis* is about 88,000 tons (dry weight), accounting for 75% of the total output of nori in China and more than 50% of the total output of nori in the world [1]. It is a nutrient-rich marine food material and a pillar product of the rural economy of China.

The nutritional value and flavor are closely related to the content of protein and amino acids. *Pyropia* is favored by people due to its characteristic taste, which results from relatively large amounts of flavor amino acids, such as glutamic acid, aspartic acid, and glycine [4]. At present, the research on flavor amino acids in *Pyropia* is mainly limited to the comparison of nutrient contents [5,6], the extraction methods [7], the taste index of food [8,9], and the factors affecting the content of amino acids. The content of flavor amino acids is mainly related to external nutrition supply and harvesting time [6,9,10]. Li et al. [10] reported that the amino acid contents could increase under the combined application of nitrogen and phosphorus fertilizers in *P. yezoensis*. However, few studies have focused on the functional characterization of key enzymes in the amino acid metabolism in red algae.

Due to its high content, glutamic acid is considered to be the main umami amino acid in *Pyropia* [6]. There are two ways to synthesize glutamic acid in organisms, one is dependent on glutamine synthetase (GS)/glutamate synthase (GOGAT), and the other is dependent on glutamate dehydrogenase (GDH) [11]. Although GS/GOGAT is considered to be the main pathway of ammonium assimilation, GDH can be used as an auxiliary approach and play an irreplaceable role in carbon and nitrogen metabolism in organisms.

The biosynthetic pathway of glutamic acid dependent on GDHs is as follows (Appendix A): glucose is converted to pyruvate through the Embden-Meyerhof-Parnas (EMP) pathway; the latter is converted to α-oxoglutarate in the tricarboxylic acid (TCA) cycle; α-oxoglutarate is catalyzed to glutamic acid by glutamate dehydrogenase (GDH) [11]. GDH is the important enzyme in glutamic acid biosynthesis, and is also one of the core enzymes in nitrogen metabolism [12]. GDHs can be divided into four distinct classes, of which GDH-1 and GDH-2 are small hexamer enzymes that are widely distributed in animal and plant tissues and play an important role in the process of ammonia assimilation [13]. GDH-3 has a large molecular weight, which acts on the catabolism of glutamate. GDH-4, discovered in eubacteria, is about 180 kDa in size and has NAD^+^ specificity [13]. According to different types of coenzymes, GDHs can be divided into the following three types: the first type uses NAD(H) as a coenzyme (EC 1.4.1.2), the second type is GDH that relies on NADP(H) as a coenzyme (EC 1.4.1.4), and the third type GDH (EC 1.4.1.3) can rely on NAD(H) or NADP(H) as a coenzyme [14]. In higher plants, GDH-catalyzed reaction is not considered to be the main pathway for glutamic acid biosynthesis, because GDHs have a relatively high *K_m_* value for ammonium ions. In contrast, GDHs in some lower organisms exhibit higher affinity for ammonium and play a more significant role in ammonium assimilation [15]. GDHs can catalyze the synthesis of glutamic acid from ammonia and α-oxoglutarate, which serve as a link between carbohydrate and amino acid metabolism [16,17]. Moreover, they play a key role in controlling glutamic acid homeostasis and supplementing the lack of carbon under certain physiological conditions [14]. GDHs also appear to be more active when energy supply is low due to abiotic stress [18,19]. They have been functionally verified in vitro in many organisms, such as *Salmonella typhimurium* and *Rumen ciliate* [20,21]. In addition, the function of GDHs has also been verified by in vivo experiments in higher plants but not in algae. The content of glutamic acid in tomato fruit transformed with the *Aspergillus GDH* gene was twice that of the control group [22]. However, the function of GDHs in *Pyropia* has not been investigated, although it is closely related to the quality traits of laver.

In this study, we cloned two *GDH* genes from the red alga *P. haitanensis*, investigated their sequence structures and enzymatic characteristics, and examined their transcription profiles under various abiotic stresses. We aim to verify the role of GDHs in the biosynthesis of glutamic acid and to provide a reference for the improvement of quality traits of *P. haitanensis*.

## 2. Results

### 2.1. Sequence Analysis of PhGDH1 and PhGDH2

The sequence features of PhGDH1 (GenBank accession: MZ614861) and PhGDH2 (GenBank accession: MZ614862) are summarized in Table 1. The ORF of PhGDH1 (1386 bp) encoded 461 amino acids, which had a predicted molecular weight (MW) of 49.30 kDa and an isoelectric point (pI) of 5.83. The ORF of PhGDH2 (1668 bp) encoded 555 amino acids, which harbored a predicted MW of 56.78 kDa and a pI of 7.10. Secondary structure prediction showed that both GDHs had similar composition, with a predominance of α-helix structure (~45%). The results of subcellular localization showed that PhGDH1 was potentially located in the cytoplasm, while PhGDH2 was presumed to be in the chloroplast.

BLAST analysis showed that PhGDH1 had 99% and 75% similarity to GDH from *Pyropia yezoensis* (GenBank accession: AAP83848.1) and *Porphyridium purpureum* (GenBank accession: KAA8497679.1), respectively (Figure 1a). PhGDH2 had 68% and 65% similarity to GDH from *Gracilariopsis chorda* (GenBank accession: PXF49351.1) and *Galdieria sulphuraria* (GenBank accession: XP_005705428.1), respectively (Figure 1b). All six GDH sequences contained the ELFV_dehydrog and ELFV_dehydrog_N domains, which reveals that they belong to the ELFV dehydrogenase family. The enzyme of this family can catalyze a reversible oxidative deamination reaction. The result of SWISS-MODEL analysis showed that PhGDH1 had 52.69% similarity to GDH from *Plasmodium falciparum* (PfGDH, 3r3j.1.A), and PhGDH2 had 42.31% similarity to human GDH (1l1f.1.A) (Appendix A). Both the two enzymes had two βαβαβ motifs (β7-α9-β8-β9-α10 and β10-α11-β11-α12-β12-α13 in PhGDH1; β6-α8-β7-β8-α9 and β9-α10-β10-α11-β11-α12 in PhGDH2), which formed a Rossmann fold for the binding of NADH (Figure 1).

### 2.2. Transcription Profiles of PhGDHs and the Content of Glutamic Acid

The trend of glutamic acid content (percentage) in *P. haitanensis* was consistent with that of *PhGDH*2 expression level but was slightly different from that of *PhGDH*1 expression level (Figure 2). However, the result showed that both genes were associated with the content of glutamic acid, suggesting that they might be involved in glutamic acid metabolism.

### 2.3. Expression and Purification of PhGDH1 and PhGDH2

To functionally characterize PhGDHs and compare their enzyme characteristics, we first induced large amounts of GDH protein expression. The MWs of PhGDH1 and PhGDH2 fusion proteins were about 70 kDa and 58 kDa, respectively. Recombinant His-tagged PhGDHs were purified, and the corresponding eluting peaks are shown in Appendix A. The Western blot (Figure 3) and SDS-PAGE (Figure 4) analyses confirmed the presence of proteins with the expected sizes.

### 2.4. Enzyme Assays and Site-Directed Mutagenesis

The purified PhGDH1 and PhGDH2 were used for the enzymatic assay. The enzyme activity was determined by measuring the variation in absorbance at 340 nm. The reaction rate in the two directions showed that the reaction rate in the direction of ammonium decomposition was much lower compared with assimilation direction (*p* < 0.05) (Appendix A). Furthermore, we have used two cofactors to detect the activity of the enzyme, and both enzymes show much higher activity against NADH than that for NADPH (Appendix A). In the following tests to determine kinetic parameters, the NADH was used as the only cofactor. The results of enzymatic characterization illustrated that the optimal reaction conditions for PhGDH1 were 25 °C and pH 8.0, and those for PhGDH2 were 25 °C and pH 8.5 (Figure 5). The calculated *K_m_* values of PhGDH1 were 0.12, 4.99, and 0.16 mM for NADH, (NH_4_)_2_SO_4_, and α-oxoglutarate, respectively; and the corresponding *K_m_* values of PhGDH2 were 0.02, 3.98, and 0.104 mM, respectively (Figure 6). The calculated *K_cat_* values of PhGDH1 were 1.52, 0.76, and 0.76 S^−1^ for NADH, (NH_4_)_2_SO_4_, and α-oxoglutarate, respectively; and the corresponding *K_cat_* values of PhGDH2 were 0.39, 0.32, and 0.32 S^−1^, respectively. The *K_cat_* values as well as *K_m_*, *V_m_* and *K_cat_/K_m_* of the PhGDH1/PhGDH2 are shown in Appendix A.

To determine the crucial active sites for PhGDHs, site-directed mutagenesis was performed (Figure 7). The catalytic activity of K137D and S293D decreased slightly compared to that of PhGDH1 (*p* > 0.05) (Figure 7a), whereas the activity of G193D and T361D decreased significantly compared to that of PhGDH2 (*p* < 0.05) (Figure 7b). The activity of G193D was 79.71% that of PHGDH2, while the activity of T361D was only 19.72%, indicating a loss of most of the activity.

### 2.5. Transcription Profiles of PhGDH1 and PhGDH2 under Abiotic Stresses

The expression of PhGDH1 and PhGDH2 showed similar tendencies under various abiotic stresses (Figure 8). Under drought stress, the expression levels of both PhGDH1 and PhGDH2 increased significantly (*p* < 0.05) (Figure 8a,b). More specifically, PhGDH1 expression reached the peak (7.5-fold) at 8 h, while PhGDH2 expression reached the peak (64-fold) at 2 h. Under high-temperature stress, the expression levels of the two genes were also increased significantly. The expression of PhGDH1 reached the maximum (39.3-fold) after 2 h of treatment, while that of PhGDH2 reached the maximum (71.3-fold) after 5 h of treatment (Figure 8c,d). The same trends were observed under ammonium salt stress. At NH_4_Cl concentration of 12 mM, both the expression level of PhGDH1 (6.1-fold) and PhGDH2 (14.9-fold) were the highest (Figure 8e,f). According to the results of qRT-PCR, both PhGDHs responded to abiotic stresses, and high temperature induced the most drastic changes in their expression levels. PhGDH2 was more sensitive to these three stresses compared to PhGDH1.

## 3. Discussion

Glutamic acid is an important flavor substance, but its metabolic pathways and relevant catalytic enzymes in red algae are scarcely studied. In this study, we measured the content of glutamic acid in *P. haitanesis* sampled from four different locations of China and found that the content of glutamic acid was higher in *P. haitanesis* from the southern region (Putian) than in that from the northern region (Yancheng). Moreover, the correlation analysis of glutamic acid content and the expression of PhGDHs showed a consistent trend, indicating that PhGDHs might be related to glutamic acid metabolism. In higher plants, the GS/GOGAT is considered to be the main pathway of ammonium assimilation. However, our unpublished data on the RNA-seq result of *P. haitanensis* samples collected from different harvesting stages showed that GS unigenes were found but with very low RPKM (Reads Per Kilobase per Million mapped reads) values (<0.5) (Appendix A). This might imply the lower activity of GS in *P. haitanensis*. Therefore, we conjected that the PhGDHs could participate in the glutamic acid biosynthetic pathway. We further identified two *GDH* genes from *P. haitanensis*, *PhGDH*1 and *PhGDH*2. They have similar domains to other GDHs from red algae, which shows that they do have the function of dehydrogenase. We compared their sequence characteristics as well as in vitro enzyme activities and aim to elucidate possible mechanisms for the flavor and stress resistance ability of *P. haitanensis*.

GDHs can be divided into four categories according to their metabolic specificity and subunit size [23], GDH-1 and GDH-2 are small hexamer enzymes, while GDH-3 and GDH-4 have a large molecular weight. In this study, both PhGDH1 and PhGDH2 are small hexameric enzymes (~50 kDa), which belong to GDH-1 or GDH-2. Generally, in hexameric GDHs, each subunit is divided into two domains, and there is a deep cleft between the two domains [24]. Domain I is mainly composed of the N-terminus of the polypeptide chain, responsible for the symmetrical binding of subunits, and participates in the formation of hexamers. Domain Ⅱ is composed of the C-terminal part of the chain and participates in the binding of the cofactor [24]. In PhGDHs, each subunit can also be divided into two domains. According to the secondary structure prediction results, both contain classic Rossmann fold for binding NAD(P)H. Both PhGDH1 and PhGDH2 can use NADH or NADPH as coenzymes, so they may belong to the third type GDH (EC 1.4.1.3). However, they show much higher activity against NADH than that for NADPH, so NADH is the main cofactor for PhGDH1 and PhGDH2. To confirm the involvement of candidate residues in the binding of NADH in *P. haitanensis*, we mutated the putative residues Lys^137^ and Ser^293^ of PhGDH1, and Gly^193^ and Thr^361^ of PhGDH2 to aspartic acid. These residues in the same position in the GDH from *Corynebacterium glutamicium* have been confirmed to be active sites [24]. All the mutated genes can express soluble proteins in *E. coli*, suggesting that none of these sites prevented the protein from folding efficiently. The activities of K137D and S293D decreased slightly; however, the G193D and T361D activities significantly decreased, which indicates that Gly^193^ and Thr^361^ are important for the binding of NADH in *P. haitanensis*. Notably, these two sites are different in GDHs from *Gracilariopsis chorda* and *Galdieria sulphuraria* (Figure 1), suggesting Gly^193^ and Thr^361^ may be novel NADH-binding sites in *P. haitanensis*.

GDHs catalyze a reversible reaction. We therefore tested the reaction rate in the two directions in vitro. The reaction rate in the direction of glutamic acid degradation was much lower (*p* < 0.05), implying the predominant role of PhGDHs catalyzing the biosynthesis of glutamic acid. In the ammonium assimilation direction, PhGDH1 and PhGDH2 had similar optimal reaction temperature and pH. Both PhGDHs exhibited the highest catalytic efficiency at 25 °C, which was close to the suitable growth temperature of *P. haitanensis* (20 °C). Their optimal reaction temperature is close to the growth temperature of *Laccaria bicolor* (30 °C) [25] and *Bacillus subtilis natto* (30 °C) [26], but lower than that of *Phormidium laminosum* (60 °C) [27] and *Pyrococcus horikoshii* (90 °C) [28]. We speculate that the optimal reaction temperature of GDHs may be related to the growth temperature specific to different organisms. The two PhGDHs are suitable to catalyze the reaction in an alkaline environment (the optimal pH values of PhGDH1 and PhGDH2 are 8.0 and 8.5, respectively), which may be related to the weak alkalinity of seawater. However, PhGDH2 is more sensitive to acidity than PhGDH1, and PhGDH2 lost most of its activity at pH 6.5. It has been previously reported that the optimal pH values for the catalytic reaction of GDHs from *Bryopsis maxima* [29], *Pyrococcus horikoshii* [28], and *Gigantocotyle explanatum* [30] are 7.5, 7.6, and 8.0, respectively. Although these GDHs possess different optimal pH values, they all exhibit higher catalytic activities in the alkaline environment.

For the three substrates, the *K_cat_* values of PhGDH1 are much higher, which means it has higher catalytic rate. Both PhGDHs had similar *K_m_* values (0.16 mM and 0.104 mM) for α-oxoglutarate, which are lower than those of GDHs from *Pyrococcus horikoshii* (K_m_ = 0.53 mM) [28] and *Thermus thermophilus* (*K_m_* = 3.5 mM) [31]. However, PhGDH2 showed a much lower *K_m_* value for NADH compared to PhGDH1, which may be due to certain differences in the cofactor-binding sites between the two enzymes. The *K_m_* value for NH_4_^+^ can reflect the ability of ammonia assimilation, and the *K_m_* values of PhGDH1 and PhGDH2 for (NH_4_)_2_SO_4_ are remarkably lower than that of GDHs in *Cucurbita pepo* (*K_m_* = 33.3 mM) for NH_4_^+^ [32]. PhGDH1 and PhGDH2 present much higher affinity for NH_4_^+^ than GDHs from most higher plants (*K_m_* = 10–80 mM) [33]. It is reasonable to speculate that they can assimilate ammonium more effectively. This phenomenon may be related to the growing environment of *P. haitanensis*, where it needs to adapt to periodic dehydration stress and consequent high-temperature stress, while the habitat of the higher plants is more stable. When under abiotic stresses, the concentration of ammonia in *P. haitanensis* cells increases. Therefore, PhGDHs with higher affinity for NH_4_^+^ is vital for the survival of *P. haitanensis*. Overall, the *K_m_* values of PhGDH2 for the three substrates were lower compared to PhGDH1, especially that for the NADH, but its *K_cat_*/*K_m_* values were also lower. Therefore, PhGDH1 has higher catalytic efficiency than PhGDH2, while PhGDH2 has a higher affinity for the three substrates.

*P. haitanensis* inhabits the intertidal zone and must undergo periodic dehydration and rehydration processes. Therefore, it has evolved a set of molecular mechanisms to cope with abiotic stresses. Many studies have reported that GDHs are involved in plant response to abiotic stresses. For example, Mena-Petite et al. [34] showed that GDH activities increased by more than 170% when *Pinus radiata* seedlings were subjected to drought stress. In addition, Tang et al. [35] observed that the GDH activity increased under heat stress in coral. In the present study, we found that abiotic stresses can significantly increase the expression levels of both PHGDH1 and PHGDH2 (*p* < 0.05). At the beginning of the abiotic stress, the expression of the two *GDH* genes rose rapidly and maintained at a high level throughout the stress compared with the control group. As the stress intensity increased, their expression levels declined, which may be due to the damage of the algal cells caused by the prolonged abiotic stress. High temperature induced the most drastic changes in the expression of *PHGDH*1 and *PHGDH*2. One of the possible reasons is that the activity of glutamate synthase decreases under high-temperature stress, which further aggravates the effect of elevated ammonium ion concentration [36]. Desiccation or high-temperature stress leads to the hydrolyzation of a large number of proteins, resulting in the increase of ammonia in algal cells. If the accumulated ammonia is not scavenged in time, it would have a toxic effect on algal cells [37]. GDHs can reduce the toxic damage resulted from excessive accumulation of NH_4_^+^ by assimilating α-oxoglutarate and NH_4_^+^ into glutamic acid during carbon metabolism. In addition, glutamic acid can continue to synthesize proline and improve the resistance of plants to abiotic stresses [38]. In this study, both PhGDHs responded to ammonium salt stress, which provides the evidence that these enzymes can synthesize glutamic acid to resist high-NH_4_^+^ stress. According to the performance of *PhGDH*1 and *PhGDH*2 under various abiotic stresses, we presume that they play a role to help *P. haitanensis* in adapting to the harsh environment of the intertidal areas.

## 4. Materials and Methods

### 4.1. Sample Collection and Treatments

*P. haitanensis* thallus was collected from different areas of China, including Putian (Fujian Province), Cangnan (Zhejiang Province), Dongtou (Zhejiang Province), and Yancheng (Jiangsu Province), from November to December 2018 (Figure 9). The sample was filtered and washed with sterile seawater twice and stored in liquid nitrogen before RNA extraction. The laver from these four locations was sent to Analysis & Detection Center, Institute of Oceanology, Chinese Academy of Sciences (Qingdao, China) for glutamic acid analysis. Prior to abiotic stress treatments, *P. haitanensis* thallus was incubated in sterile seawater at 20 °C for more than 24 h. Variable-to-maximum fluorescence (*F_v_*/*F_m_* = (*F_m_* − *F_0_*)/*F_m_*) was measured using a FC1000-H fluorescence imaging system (Photon Systems Instruments, Czech Republic) to determine the growth status of the thallus (Appendix A). For dehydration treatments, the samples were placed in a dry petri dish in the darkness for 2, 4, 6, 8, and 10 h at 20 °C; for high-temperature stress, the samples were placed in the darkness at 30 °C for 1, 2, 3, 4, and 5 h; the samples were placed in 12, 60, and 300 mM NH_4_Cl for 2 h in the darkness at 20 °C for ammonium salt stress.

### 4.2. Cloning and Sequence Analysis of PhGDH1 and PhGDH2

Total RNA was extracted with the Plant RNA Kit (OMEGA, China) and converted into cDNA with the Transcriptor First Strand cDNA Synthesis Kit (Takara, Japan) according to the manufacturers’ instructions. Sequences annotated as GDH in the transcriptome of *P. haitanensis* (accession: PRJNA428906, accessed on 8 January 2018) were BLAST against the NCBI nucleotide database, and then two *GDH* sequences (*PhGDH*1 and *PhGDH*2) with the highest identities were selected. The PhGDH coding sequences are listed in Appendix A. The open reading frames (ORFs) of PhGDH1 and PhGDH2 were amplified with primers of PhGDH1-F, PhGDH1-R, PhGDH2-F, and PhGDH2-R (Appendix A) and 2× Phanta Master Mix (Vazyme, China). PCR program was as follows: 98 °C for 5 min; 35 cycles of 98 °C for 10 s, 55 °C for 15 s, and 72 °C for 90 s; and 72 °C for 10 min.

The obtained *PhGDH*1 and *PhGDH*2 coding sequences were translated into amino acid sequences with ORF Finder [39], which were then aligned with other GDH proteins by CLUSTALW (https://www.genome.jp/tools-bin/clustalw, accessed on 25 February 2021). The physical and chemical parameters (molecular weight, isoelectric point) of PhGDH1 and PhGDH2 were predicted with ProtParam [40], and the motifs of PhGDH1 and PhGDH2 were analyzed by the MOTIF tool (http://www.genome.jp/tools/motif/, accessed on 25 February 2021). The subcellular localization of PhGDHs was predicted by TargetP v1.1 [41], and the SignalP v4.1 Server (http://www.cbs.dtu.dk/services/SignalP-4.1/, accessed on 25 February 2021) was used to predict signal peptides [41]. The transmembrane helices were predicted with the TMHMM Server v2.0 (http://www.cbs.dtu.dk/services/TMHMM/, accessed on 25 February 2021). The tertiary structures of PhGDH1 and PhGDH2 were predicted by SWISS-MODEL [42], and the secondary structures were illustrated by ESPript [43].

### 4.3. Expression and Purification of PhGDH1 and PhGDH2

The pET and pCold systems were used to express PhGDH1 and PhGDH2 in vitro, respectively. The full-length ORF of *PhGDH*1/*PhGDH*2, which was amplified as an *Eco*R I/*Hind* III fragment by PCR, was cloned into the vectors pET-32a and pCold-I with His-tagged. PCR program was as follows: 98 °C for 5 min; 35 cycles of 98 °C for 10 s, 55 °C for 15 s, and 72 °C for 90 s; and 72 °C for 10 min. The *Escherichia coli* cells (BL21 (DE3) pLysS and Transetta (DE3)) were transformed with the recombinant expression plasmids (pET-PhGDH1 and pCold-PhGDH2). The transformed *E. coli* cells were then incubated in 1 L of Luria–Bertani (LB) medium with 100 μg·mL^–1^ of ampicillin and 20 μg·mL^–1^ of chloramphenicol at 37 °C. When OD600 reached 0.6, 0.1 mM IPTG was supplemented, and the inducement conditions were changed to 15 °C and 120 rpm. After 24 h of culturing, the transformed cells were harvested by centrifugation at 4500 rpm and 4 °C for 30 min. Before large-scale purification, the cells were disrupted by an ultrasound cell crusher (Xinzhi, Ningbo, China). One cOmplete, EDTA-free tablet (Roche, Switzerland) was added to the 35 mL extracts before centrifugation at 12,000 rpm and 4 °C for 45 min.

The overexpressed PhGDH1 and PhGDH2 were purified by Ni^2+^-affinity chromatography using the ÄKTA Pure system (GE Healthcare, Fairfield, CA, USA) equipped with a His HP (GE Healthcare, Fairfield, UK). The column was equilibrated with 50 mL (10 column volumes) of buffer A (20 mM sodium phosphate, 20 mM imidazole, 500 mM NaCl, and 5% glycerin; pH 8.0) at a flow rate of 5 mL·min^–1^. Then 150 mL sample (bacterial extract diluted 5 times by buffer A) was injected at a rate of 1.2 mL·min^–1^. The protein was eluted by a gradient increase in the proportion of buffer B (20 mM sodium phosphate, 500 mM imidazole, 500 mM NaCl, and 5% glycerin; pH 8.0) at a rate of 3 mL·min^–1^. When the proportion of buffer B increased to 15%, a large number of recombinant proteins were eluted. The elution was tested for the presence of the target protein by Western blot using Anti His-Tag mouse monoclonal antibody and Goat anti-mouse IgG (HRP conjugated) (CWBIO, Beijing, China), after separated by 12% sodium dodecyl sulphate-polyacrylamide gel electrophoresis (SDS-PAGE). The concentration of protein was determined with the BCA Protein Assay Kit (Solarbio, Beijing, China).

### 4.4. Enzyme Assays

The enzyme activity in the direction of glutamic acid degradation of PhGDH1 and PhGDH2 was measured by NADH-glutamate dehydrogenase (NADH-GDH) kit (Grace Biotechnolgy, Suzhou, China). The enzyme activity in the assimilation direction of PhGDH1 and PhGDH2 was measured according to Thatcher and Storey [44], with some modifications. Enzyme assays were conducted by monitoring the change of OD_340_. Standard reaction mixtures for PhGDH1/PhGDH2 assays contained 632.5 μL of sodium phosphate buffer, 37.5 μL of 0.1 M α-oxoglutarate, 37.5 μL of 1 M (NH_4_)_2_SO_4_, 5 μL of 18 mM ADP, 25 μL of 10 mM NADH/NADPH, and 25 μL of purified PhGDH1/PhGDH2. One unit of enzyme activity is defined as catalyzing 1.0 μM NADH/NADPH to NAD^+^/NADP^+^ per minute. To obtain the optimal reaction conditions for the enzyme, we measured the enzyme activity at various temperatures (15, 20, 25, 30, 35, and 40 °C) and different pH values (6.5, 7.0, 7.5, 8.0, 8.5, 9.0, and 9.5). To obtain the kinetic constant *K_m_* of PhGDH1 and PhGDH2, the reaction rate was measured at different concentrations of NADH (0.1, 0.2, 0.3, 0.4, 0.5, and 0.6 mM), (NH_4_)_2_SO_4_ (10, 20, 30, 40, 50, and 60 mM), and α-oxoglutarate (1, 2, 3, 4, 5, and 6 mM) under optimal conditions determined, and the *K_m_*, *V_m_* and *K_cat_* values were calculated by the double reciprocal plot method [45].

### 4.5. Site-Directed Mutagenesis

Site-directed mutagenesis was conducted using the Mut Express II Fast Mutagenesis Kit V2 (Vazyme, Nanjing, China). The putative residues, Lys^137^ and Ser^293^ of PhGDH1, and Gly^193^ and Thr^361^ of PhGDH2, were mutated to aspartic acid referred to the GDH crucial residues study by Son et al. (2015). The four pairs of primers (K137D-F/R, S293D-F/R, G193D-F/R, and T361D-F/R) were selected to introduce base substitutions (Appendix A). PCR program was as follows: 95 °C for 30 s; 30 cycles of 95 °C for 15 s, 64 °C for 15 s, and 72 °C for 6 min; and 72 °C for 5 min. The pET-PhGDH1/pCold-PhGDH2 was used as the template, and the *E. coli* cells (BL21 (DE3) pLysS and Transetta (DE3)) were transformed with the products. The purified mutant protein was harvested by the same method as described for harvesting PhGDH1/PhGDH2. The catalytic activity of the mutant protein was detected by the GDH activity assay kit (MLBIO, Shanghai, China).

### 4.6. Quantitative Real-Time PCR (qRT-PCR) Analysis

The primers (qPhGDH1-F, qPhGDH1-R, qPhGDH2-F, and qPhGDH2-R) were selected for qRT-PCR analysis (Appendix A) and the EF2 gene was used as an internal control [46]. The qRT-PCR was performed with 2× SYBR Green qPCR Mix (SparkJade, Shandong, China) on a TP800 Thermal Cycler Dice (Takara, Otsu, Japan). The protocol was 94 °C for 3 min; 40 cycles of 94 °C for 20 s, 50 °C for 20 s, and 72 °C for 30 s. Three biological replicates were performed. The 2^−^^△△Ct^ method was used to calculate the relative quantitative value, and SPSS 26.0 was used for statistical analysis based on the ΔCt values.

## 5. Conclusions

In this study, PhGDH1 and PhGDH2 were verified to catalyze the biosynthesis of glutamic acid from ammonia and α-oxoglutarate with relatively lower *K_m_* values compared with those in higher plants. They showed higher affinity for NH_4_^+^, indicating that they can assimilate ammonium more effectively than higher plants. Their expression levels were significantly increased under various abiotic stresses, implying that PhGDHs may play an important role in *P. haitanensis* adapting to the harsh environment of the intertidal areas. The study of PhGDHs provides a better understanding of the glutamic acid biosynthetic pathway in *P. haitanensis* and lays foundation for improving the quality of *P. haitanensis*.

## Figures and Tables

**Figure 1 molecules-26-06793-f001:**
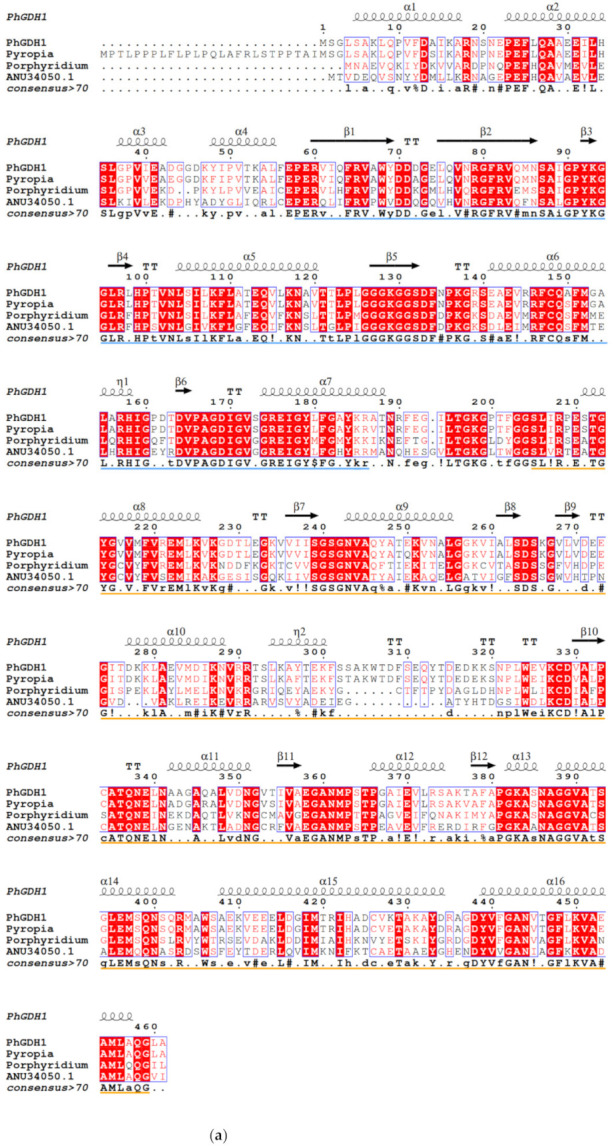
Multiple sequence alignment of PhGDH1 (**a**) and PhGDH2 (**b**), respectively, with GDHs from representative species. ELFV_dehydrog_N and ELFV_dehydrog domains are marked with blue and yellow underlines, respectively.

**Figure 2 molecules-26-06793-f002:**
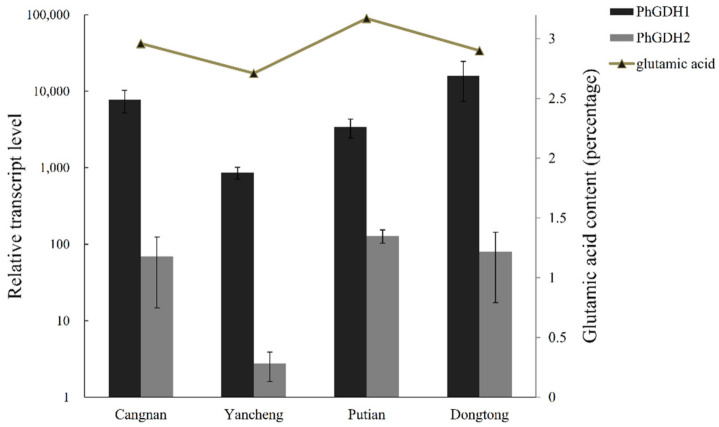
Transcription profiles of *PhGDH*1 and *PhGDH*2, as well as glutamic acid contents in *P. haitanensis* sampled from different cities. “percentage” means the weight ratio of amino acids to the dry weight of *P. haitanensis*.

**Figure 3 molecules-26-06793-f003:**
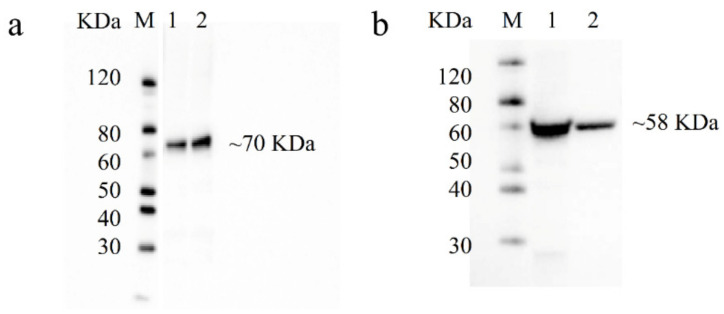
Western blot analysis of recombinant PhGDH1 (**a**) and PhGDH2 (**b**). M: protein ladder; 1–2: purified PhGDH1/PhGDH2 fusion proteins.

**Figure 4 molecules-26-06793-f004:**
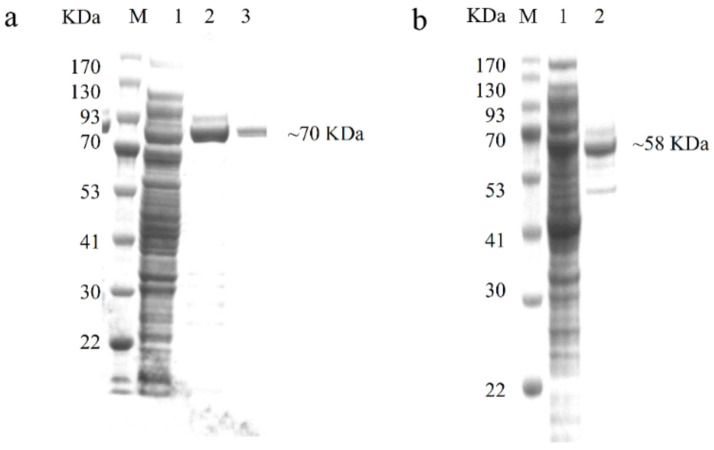
SDS-PAGE results of recombinant PhGDH1 and PhGDH2. (**a**): Expression analysis of recombinant PhGDH1, M: protein ladder, 1: Expression of PhGDH1 induced with 0.1 mM IPTG, 2–3: purified PhGDH1 fusion protein. (**b**): Expression analysis of recombinant PhGDH2, M: protein ladder, 1: Expression of PhGDH2 induced with 0.1 mM IPTG, 2: purified PhGDH2 fusion protein.

**Figure 5 molecules-26-06793-f005:**
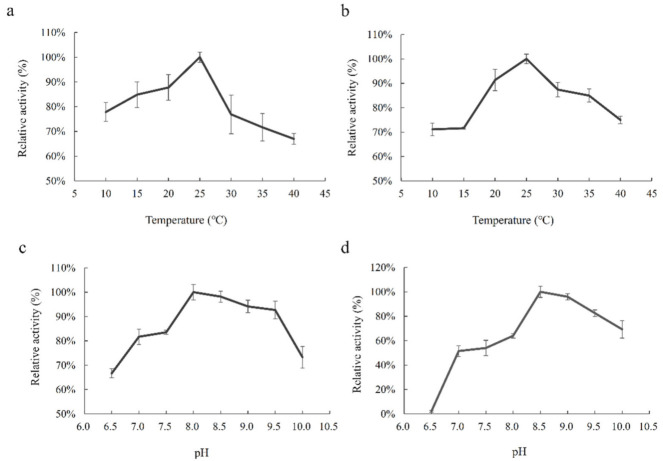
Influence of temperature and pH on the activities of PhGDH1 and PhGDH2. Influence of temperature (10–40 °C) on the activity of PhGDH1 (**a**) and PhGDH2 (**b**). Influence of pH (6.5–10.0) on the activity of PhGDH1 (**c**) and PhGDH2 (**d**).

**Figure 6 molecules-26-06793-f006:**
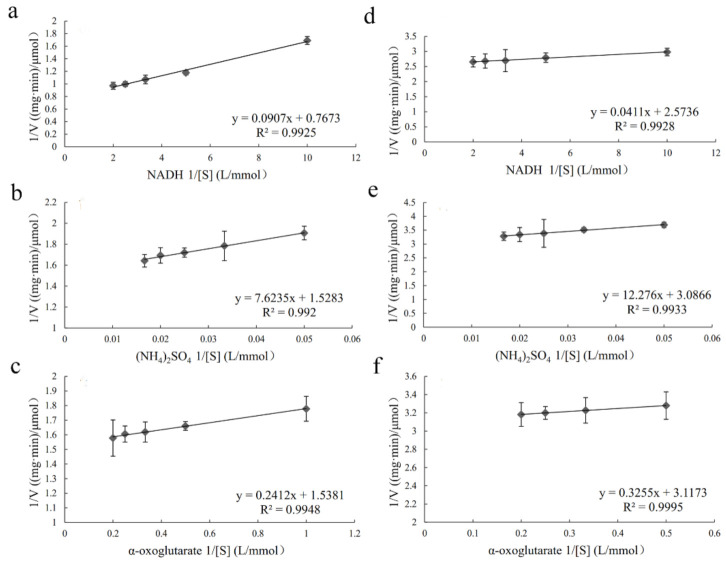
Kinetic analysis of PhGDH1 and PhGDH2. The *K_m_* values of PhGDH1 for the substrates of NADH (**a**), (NH_4_)_2_SO_4_ (**b**), and α-oxoglutarate (**c**). The *K_m_* values of PhGDH2 for the substrates of NADH (**d**), (NH_4_)_2_SO_4_ (**e**), and α-oxoglutarate (**f**).

**Figure 7 molecules-26-06793-f007:**
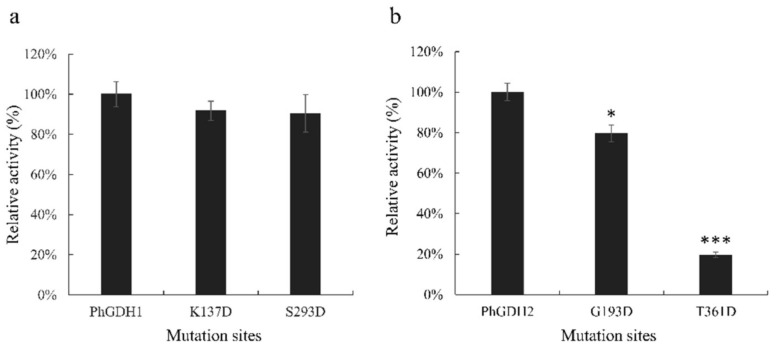
Site-directed mutagenesis. (**a**): Comparisons of the relative activities between recombinant mutant PhGDH1 proteins and wild-type PhGDH1. (**b**): Comparisons of the relative activities between recombinant mutant PhGDH2 and wild-type PhGDH2. Residues involved in the stabilization of the cofactor were replaced by appropriate residues. * *p* < 0.05 and *** *p* < 0.001.

**Figure 8 molecules-26-06793-f008:**
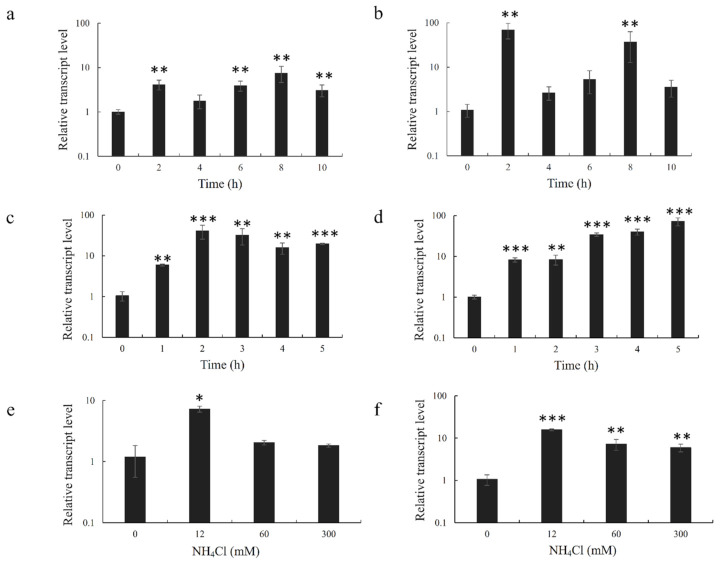
Transcription profiles of *PhGDH*1 and *PhGDH*2 under drought stress (**a**,**b**), high-temperature stress (**c**,**d**), and ammonium salt stress (**e**,**f**). * *p* < 0.05, ** *p* < 0.01, and *** *p* < 0.001.

**Figure 9 molecules-26-06793-f009:**
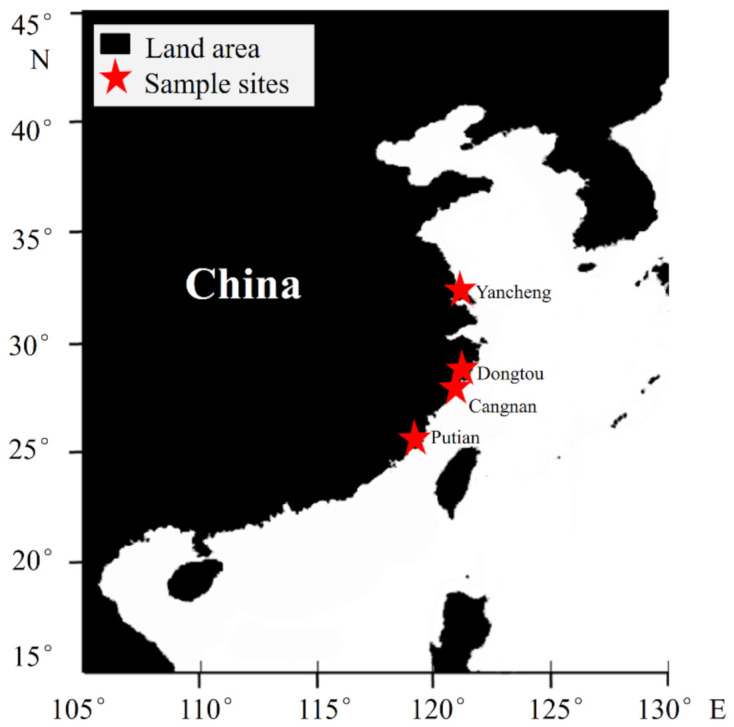
The sampling sites of *P. haitanensis*. The thallus was collected from Putian (25°28′ N, 119°02′ E), Dongtou (27°51′ N, 121°08′ E), Cangnan (27°30′ N, 120°24′ E), and Yancheng (33°24′ N, 120°09′ E).

**Table 1 molecules-26-06793-t001:** Comparisons of PhGDH1 and PhGDH2 in terms of gene/protein structure, subcellular localization, and activities. AA, amino acids; CDS, coding sequences; OT, other; cTP, chloroplast transit peptides.

Name	PhGDH1	PhGDH2
Length of CDS (bp)	1386	1668
Size of protein (AA)	461	555
Molecular weight (KDa)	49.30	56.78
Isoelectric point	5.83	7.10
α-helix (%)	45.55	44.50
Extened strand (%)	12.80	12.43
β-turn (%)	9.33	8.83
Random coil (%)	32.32	34.23
Signal peptide	0	0
Subcellular location	OT	cTP
Transmembrane helices	0	0
Conserved domain	ELFV_dehydrog; ELFV_dehydrog_N	ELFV_dehydrog; ELFV_dehydrog_N

## Data Availability

Not applicable.

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
