# Peer review of "Glutamate Dehydrogenase Functions in Glutamic Acid Metabolism and Stress Resistance in Pyropia haitanensis"

_molecules, 2021, doi:10.3390/molecules26226793_

Round 1

Reviewer 1 Report

There are a few comments that need to be addressed before accepting the article for publication. They are as follows:

  1. Line 113: "ELFV dehydrogenase" add the definition of the family for better understanding.
  2. Figure 3: what are the units on the X-axis of figure 3a and 3b? Extract the raw data and make plots using standard graph applications (like excel/origin/sigma plot) for better visibility.
  3.   Line 229: What is RPKM? It is not defined anywhere in the manuscript?
  4. Line 238-256: Reconsider writing this paragraph again for flow. The authors stated 'four categories of GDHs'; however, they talked only about two types. What are the other two types?

Reviewer 2 Report

Introduction:

Line 29, make it simple, Pyropia is an important genus of red algae.

The sentence in Line 49 and Line 52 seems irrelevant. I do not see any relation between these sentences.

What class is the GDH? Please mention the class of the enzyme and EC number in the introduction. Also described the common features of the enzyme (eg intra or extracellular)

Overall, the introduction is well written.

Results:

Have you determined the conserved region of the sequence? If yes, please mention the conserved region (amino acid residues) when compare to other gdh genes.

Are these enzymes required cofactor? Have you determined which residues that act as cofactor-binding site?

Line 129, in the manuscript, the authors say that the expression level was slightly different. What do you mean by this, because according to your data, the expression level of pHGDH2 is significant compare to that of pHGDH1, right? Have you analyzed the data with statistic so you came up with a conclusion that the expression level was slightly different?

The drawback of this study that the author using online SWISS Model to predict the structure, probably in the future this research group manage to resolve the crystal structure of the enzyme.

According to your statements that the enzymes play important role in glutamic acid synthesis, so in this case, have you ever tried to determine the concentration of glutamic acid in the mixtures? Or have you ever try to do the conversion reaction?

Line 229, what is RPKM stand for? Please describe briefly what if the value of RPKM is low or high

Conclusion: I think the author have not cover all the results in the conclusion.

Materials and method

Table 4 is unnecessary; you can add in the supplementary material.

Line 372 please correct “pCold-1 tagged with His (6)” to pCold-1 with His-tagged

Why PhGDH 2 is clone to pCold-1? Is this protein required low temperature to fold properly?

Line 381 please mention the concentration of the protease inhibitor that you add during the cell lysis.

Line 403, what is the concentration and the pH of sodium phosphate buffer?.

In the methods, you have measured the optimum temp and optimum pH in the range of 15-40oC and 6.5-9.5 respectively. Have you tried at higher temperatures (>40oC) and at more acidic conditions (pH <5.0).

Have you determined the stability of the enzyme? In my opinion, optimum temp and pH cannot be used for the conversion reaction, it’s a different thing.

Please underlined the restriction sites if you introduced the restriction sites in the primers.

Please mention the PCR conditions for the genes prior to cloning.
